# Design, Construction and Control of a Manipulator Driven by Pneumatic Artificial Muscles

**DOI:** 10.3390/s23020776

**Published:** 2023-01-10

**Authors:** Željko Šitum, Srečko Herceg, Nenad Bolf, Željka Ujević Andrijić

**Affiliations:** 1Department of Robotics and Production System Automation, Faculty of Mechanical Engineering and Naval Architecture, University of Zagreb, Ivana Lučića 5, 10000 Zagreb, Croatia; 2Department of Measurements and Process Control, Faculty of Chemical Engineering and Technology, University of Zagreb, Savska cesta 16/5a, 10000 Zagreb, Croatia

**Keywords:** pneumatic artificial muscle (PAM), manipulator, control, experimental system

## Abstract

This paper describes the design, construction and experimental testing of a single-joint manipulator arm actuated by pneumatic artificial muscles (PAMs) for the tasks of transporting and sorting work pieces. An antagonistic muscle pair is used in a rotational sense to produce a required torque on a pulley. The concept, operating principle and elementary properties of pneumatic muscle actuators are explained. Different conceptions of the system realizations are analyzed using the morphological-matrix conceptual design framework and top-rated solution was practically realized. A simplified, control-oriented mathematical model of the manipulator arm driven by PAMs and controlled with a proportional control valve is derived. The model is then used for a controller design process. Fluidic muscles have great potential for industrial applications and assembly automation to actuate new types of robots and manipulators. Their characteristics, such as compactness, high strength, high power-to-weight ratio, inherent safety and simplicity, are worthy features for advanced manipulation systems. The experiments were carried out on a practically realized manipulator actuated by a pair of muscle actuators set into an antagonistic configuration. The setup also includes an original solution for the subsystem to add work pieces in the working space of the manipulator.

## 1. Introduction

Technological improvements and innovations in modern pneumatic components as well as in control strategies have made some new modalities in traditional pneumatic-system applications possible. One of the research directions relates to the area of pneumatic artificial-muscle actuators (PAMs), which use biological principles for system development and control, and attempt to mimic natural human movement. Some new applications are also being identified, particularly in bio-robotics and in human-friendly orthopedic aids for the rehabilitation of polio patients. PAMs are increasingly being explored and used in advanced human-like robotic systems, many of which have natural compliance properties. Fluidic muscles also have great potential for industrial applications and assembly automation to actuate new types of robots and manipulators. Their characteristics of compactness, high strength, high power-to-weight ratio, inherent safety and simplicity are valuable features for advanced manipulation systems. There are several types of fluidic muscles based on the use of rubber or similar elastic materials, such as McKibben’s artificial muscle [1,2,3], Bridgestone’s rubbertuator [4,5], Shadow Robot Company’s air muscle [6], Festo’s fluidic muscle [7,8], the folded PAM developed by the Vrije University of Brussels [9], the ROMAC (RObotic Muscle ACtuator), Yarlott and Kukolj’s PAM [10] and several others. In most cases, the structure of the fluidic muscle consists of an airtight inner polymer tube contained in a flexible piece of hollow-braided construction and corresponding metal end pieces for external attachment and pressurization. When the inner membrane is inflated with compressed air, the pressurized gas pushes against the outer sheath, increasing the volume. The muscle radius increases and, together with the radial expansion, the muscle contracts axially and exerts a tensile force. Due to the highly nonlinear and time-varying nature of pneumatic muscles, it is difficult to control their force or motion. Various control methods have been used to control different robotic arms and manipulators driven by artificial muscles. The early control strategies were based on classical linear controllers [11], and then some modern control schemes were developed using adaptive controllers [12,13], sliding-mode controllers [14], fuzzy controllers [15], neural-network controllers [16] and others. In most research studies, proportional directional control valves were used. Some studies have used proportional pressure valves [17] or high-speed on/off solenoid valves controlled by a pulse-width modulated control signal [18].

This paper is a continuation of the conference paper [19] with the extension of the originally developed PAM-driven manipulator by adding a feeder and a gripper that forms a unique sorting system. The paper also presents the development of an algorithm for the optimal control of the sorting system. In this work, we used fluidic muscles in an antagonistic pair to generate the torque required to move a planar manipulator arm. The pair of PAM actuators in an antagonistic configuration imitate a biceps–triceps system and highlights the analogy between this artificial muscle and human skeletal muscle. The article begins with a brief comparison of the principle of operation of the human arm and the manipulator arm driven by two antagonistic muscle actuators. Then, an analysis of different variants of the system design for feeding, transferring and sorting workpieces with a manipulator actuated by PAMs is given. Then, a simplified control-oriented mathematical model is described. Finally, a brief description of the designed system is presented and experiments are performed on a practically realized PAM-driven manipulator.

## 2. Materials and Methods

### 2.1. Manipulator Arm Actuated by Pneumatic Artificial Muscles

The pneumatic muscle is a flexible traction actuator powered by gas pressure. It consists of an inflatable and flexible rubber tube and two connecting flanges along which a mechanical load is pulled. PAM operates at positive pressure and contracts when pressurized, generating a load-bearing capacity at its ends during this contraction, as can be seen in Figure 1. The compressed gas-powered artificial muscle actuator contracts longitudinally when stretched radially, converting the radial expansion force into an axial contraction force. PAMs are contractile devices that can generate translational and unidirectional motion. Normally, the maximum contraction of muscle actuators is about 25% of the nominal length.

To move the load in both directions, an antagonistic coupling should be used. Two antagonistic PAMs can drive a rotational element in a similar manner to how the skeletal biceps and triceps rotate the forearm around the elbow, as shown in Figure 2a. Figure 2b shows a schematic diagram of the control of a single-joint manipulator arm actuated by a bi-muscular drive system.

The initial free length of the muscles is given as *l*_0_. In the initial state, both muscles are inflated with approximately the same pressure, and when the pressures in muscles 1 and 2 are changed, length *l* of the muscles also changes, causing arm rotation by angle *θ*. Manipulator arm rotation is achieved by increasing the pressure in the agonist muscle by *Δp* and simultaneously decreasing the pressure in the antagonist muscle by the same value. The tensile force in each muscle is proportional to the control pressure, and the difference between the two forces produces a torque that rotates the pulley via a transmission belt. The joint angle depends on the difference between the forces exerted. Both actuators are initially inflated with approximately equal pressure and contracted with an initial contraction ratio. This defines the rest position of the manipulator arm. As the input pressure to one muscle (biceps) increases, the input pressure to another muscle (triceps) decreases and vice versa, creating torque at the pivot joint. A proportional control valve connected to both air muscles was used as the control component for the rotational motion control of the manipulator. The control signal changes the pressure in the muscles, resulting in a change in the length of the muscles (one muscle is shortened, the second is lengthened by approximately the same amount), and the resulting torque rotates the manipulator arm.

### 2.2. Construction Concepts for the System Realization

The functional structure of a product must clearly show the goal of product development and represents a meaningful and compatible combination of individual functions that form the overall function of the system. The relationship between the individual functions must be carefully defined in terms of the conversion of materials, energy and information. The functional structure of the system for sorting with specified input and output of energy, materials and information is shown in Figure 3.

Several variants of possible solutions to the system for sorting are generated using a morphological matrix, as shown in Figure 4.

In the process of generating system design concepts, the goal is to develop as many ideas as possible. It is important to cover the requirements and preliminary specifications of products, as well as functional decomposition and possible solutions for each function of the product. Figure 5 shows four variants that are considered in the process of the system design.

In the first variant, the task of capturing objects is realized using a parallel gripper that is moving forward with a pneumatic cylinder and allows the manipulator arm to capture a stationary object. Delivery of items in this variant is simple. The second variant shows the capture of items using a radial gripper with deflection angle of 180° that allows an easy access to stationary objects and their capture. Delivery of work items is also easy. The third option shows the capture of objects using a parallel gripper where the keeper is modified and contains a movable barrier with mechanical activation. When the barrier is moved, then item falls between the clamps. If the gripper with the object is moved, the barrier returns because of spring, thus blocking the fall of next object. Finally, in the fourth variant, objects come by conveyor-belt and are taken in a way in which an additional pneumatic muscle provides lifting and lowering of the manipulator arm, which enables capturing of items.

Selection process of the final solution, based on comparisons of different system designs, is shown in Table 1.

Considered variants are evaluated through their key parts. The criteria for evaluation were both cost and complexity of individual-part construction (P—price, C—complexity of construction). Total reliability in the work of individual solution is also estimated. Ratings are from 1 to 5, where the minimum cost, minimum complexity in construction and maximum reliability will be given priority during the selection (grade 1).

Figure 6 shows the final CAD model of the chosen system for sorting task, which includes the manipulator with gripper, the work pieces feeder and the manner of storage of the items.

After defining the main parts of the system, particular attention was given to the development of subsystems for supplying objects, which is shown in Figure 7.

The feeder is a simple mechanical assembly which is composed of several parts. The housing is robust and ensures stability of structures. The manipulator arm with a gripper achieves mechanical activation of the keeper, which moves and then allows object to fall between the fingers of end-effector when the operation of transferring the work piece starts.

### 2.3. System Modelling and Control

The pneumatic muscle is a diaphragmatic contraction drive that shortens under pressure. The single-joint manipulator arm actuated by PAMs is usually shown as an antagonistic configuration: the driving muscle is referred to as the agonist or flexor, while the braking muscle is referred to as the antagonist or extensor.

The equation of motion for the angle *θ* of the manipulator arm joint can be rewritten in standard form as follows:(1)$$\ddot{\theta }=\frac{1}{I}\;\;\left(F_{1}-\;F_{2}\right)\;\cdot r$$
where *I* is the moment of inertia of the pulley, *F*_1_ and *F*_2_ are the tensile forces, and *r* is the radius of the joint. Note that the friction of the pulley is neglected because a low-friction bearing is used in the system. Due to the specific structure of PAMs and their significant nonlinear phenomena, many researchers have studied the relationship between the force, applied air pressure, length and contraction ratio to derive a suitable expression for the equation of the force generated by PAMs. Various designs of PAM have been developed in the literature and several static and dynamic models have been proposed [2,20,21]. The most commonly used expression is a static force model, which specifies the force as a function of air pressure and contraction rate. The difference between the models is that a number of parameters are used to obtain the PAM tensile force. However, with the help of some relationships, these models can be transformed into each other [11]. In general, the muscle is considered as a cylinder with variable diameter and length. The theoretical model of PAM is based on the theory of input and output work (‘virtual work’), assuming that there are no energy losses due to material deformation, friction, etc. [22]. Equation (2) expresses the force *F* generated by the muscle at a given pressure *p*, and the muscle contraction ratio *ε* is given in the following form [2]:(2)$$F(p,\varepsilon )=p\cdot \frac{D_{0}^{2}\cdot \pi }{4}\;\left[\;a\cdot {\left(1-\varepsilon \right)}^{2}-b\;\right]$$
where $a=\frac{3}{\tan^{2}\left(\alpha_{0}\right)}$, $b=\frac{1}{\sin^{2}\left(\alpha_{0}\right)}$, $\varepsilon =\frac{L_{0}-L}{L_{0}}$, *L*_0_ and *D*_0_ are, respectively, the initial length and nominal diameter of the muscle when it is not contracting, and *α*_0_ is the initial angle of the braided sheath between the filament and the longitudinal axis of the muscle. In an equilibrium position, both actuators are inflated with an approximately equal pressure *p*_0_ and contracted with an initial contraction ratio *ε*_0_. To perform a rotational movement through a certain angle *θ*, the muscles are activated so that one of them is inflated to $p_{A}=p_{0}+\Delta p$, the contraction rate becomes $\varepsilon_{A}=\varepsilon_{0}+\Delta \varepsilon$, the length becomes $L_{0}-\Delta L$, and the pulling force becomes *F*_1_. The other one is simultaneously deflated to $p_{B}=p_{0}-\Delta p$, the contraction rate becomes $\varepsilon_{B}=\varepsilon_{0}-\Delta \varepsilon$, the length becomes $L_{0}+\Delta L$, and the contraction force becomes *F*_2_. The greater the internal pressure, the more the muscle shortens. So, *Δp* is not the difference between the two muscles but the incremental set pressure or the distance from the initial pressure *p*_0_. During the operation of the system, one muscle shortens while the second expands by approximately the same amount, guaranteeing high stiffness of the beam.

The contraction forces of the muscles can now be expressed as follows:(3)$$\begin{array}{c} F_{1}=\left(p_{0}+\Delta p\right)\;\;{\left[\;a_{1}\cdot {\left(1-\varepsilon_{A}\right)}^{2}-b_{1}\right]}^{}\\ F_{2}=\left(p_{0}-\Delta p\right)\;\;{\left[\;a_{1}\cdot {\left(1-\varepsilon_{B}\right)}^{2}-b_{1}\right]}^{} \end{array}$$
where the contraction rates are:(4)$$\begin{array}{c} \varepsilon_{A}=\varepsilon_{0}+\Delta \varepsilon \;=\varepsilon_{0}+\frac{\Delta L}{L_{0}}=\varepsilon_{0}+\frac{d\cdot \theta }{L_{0}}\\ \varepsilon_{B}=\varepsilon_{0}-\Delta \varepsilon \;=\varepsilon_{0}-\frac{\Delta L}{L_{0}}=\varepsilon_{0}-\frac{d\cdot \theta }{L_{0}} \end{array}$$
while $a_{1}=\frac{3\cdot D_{0}^{2}\;\pi }{4\cdot \tan^{2}\left(\alpha_{0}\right)}\;$ and $b_{1}=\frac{D_{0}^{2}\;\pi }{4\cdot \sin^{2}\left(\alpha_{0}\right)}\;$.

Using the expressions for the contracting forces, the actuating torque on the arm is given by:(5)$$\;\;T=\left\{\;\left(p_{0}+\Delta p\right)\left[\;a_{1}{\left(1-\varepsilon_{0}-\frac{d\cdot \theta }{L_{0}}\right)}^{2}-b_{1}\;\right]-\left(p_{0}-\Delta p\right)\left[\;a_{1}\;{\left(1-\varepsilon_{0}+\frac{d\cdot \theta }{L_{0}}\right)}^{2}-b_{1}\;\right]\;\right\}\cdot d\;\;$$

In the above expression, it is assumed that the generated torque *T* is a function of the incremental pressure *Δp* and the angular position of the beam *θ*.

The nonlinear function of the actuating torque given in a generalized form *T*(*p*, *θ*) can be linearized and expressed in the simplified form:(6)$$T=\frac{\partial T}{\partial \;\Delta p}\Delta p\;\;+\;\;\frac{\partial T}{\partial \;\theta }\Delta \theta$$

If we assume that the torque *T* only depends on the incremental pressure *Δp* and that it is not a function of the angular position of the manipulator *θ* (for small deflection angles *θ*, the counteracting effect, which explains the natural compliance of the antagonistic PAM arrangement, is ignored), then the partial derivative is:(7)$$K_{1}=\frac{\partial T}{\partial \;\Delta p}=2\cdot d\cdot \left[\;a_{1}\;{\left(1-\varepsilon_{0}\right)}^{2}-b_{1}\;\right]$$
and the simplified expression for the generated torque is then given by:(8)$$T=K_{1}\;\cdot \Delta p$$

Due to the highly nonlinear effects in pneumatic systems caused by phenomena mainly related to the compressibility of air, and due to the associated analytical complexity, it is a very challenging task to obtain a dynamic model for the pressure in the pneumatic muscle actuated by a proportional valve. Therefore, the dynamic model is obtained as the relationship between the pressure in the actuator and the control signal at the valve from the experimentally measured pressure response to a step command input. The transient pressure response has a (quasi-)aperiodic form and can be approximated by a first-order delay term, where the input is the control command *u* and the output is the pressure variance *Δp*:(9)$$\tau_{m}\cdot \Delta \dot{p}\;\;+\;\;\Delta p=K_{m}\cdot u$$
where *K*_m_ is the transfer gain and *τ*_m_ is the time constant. From the reference step change illustrated in Figure 8, the time constant of the process can be approximated numerically to the value *τ*_m_ ≈ 0.05 s in the case of pressure increasing, and *τ*_m_ ≈ 0.1 s in the case of pressure decreasing. The forward gain *K*_m_ is estimated numerically to the value *K*_m_ ≈ 1.1 × 10^5^ Pa/V. These values are used in the controller design procedure.

### 2.4. State-Space Model of the System

Using the expressions (1), (8) and (9), the mathematical model of the manipulator arm actuated by antagonistic artificial-muscle arrangement controlled by the proportional valve can be given in the following linearized form:(10)$$\begin{array}{c} \ddot{\theta }=\frac{1}{I}\;\;K_{1}\;\;\Delta p\\ \Delta \dot{p}=-\frac{1}{\tau_{m}}\;\;\Delta p\;\;+\;\;\frac{K_{m}}{\tau_{m}}\;\;u \end{array}$$

For controller design, the dynamical model of the manipulator arm is rewritten in the form of state equations. The state-variable vector *x* is chosen as follows:(11)$$x={\left[\;\begin{array}{ccc} x_{1} & x_{2} & x_{3}\; \end{array}\right]}^{\;\;T}={\left[\begin{array}{ccc} \;\theta & \dot{\theta } & \Delta p\; \end{array}\right]}^{\;\;T}$$

Then, the following linearized state-space model is obtained:(12)$$\begin{array}{c} \dot{x}=A\;x+B\;u\\ y=C\;x+D\;u \end{array}$$
where the state-space matrices are:$$A=\;\;\;\left[\begin{array}{ccc} 0 & 1 & 0\\ 0 & 0 & K_{1}/I\\ 0 & 0 & -1/\tau_{m} \end{array}\right],\;\;B=\;{\left[\begin{array}{ccc} 0 & 0 & K_{m}/\tau_{m} \end{array}\right]}^{\;\;T}\;,\;\;C=\;\;\left[\begin{array}{ccc} 1 & 0 & 0 \end{array}\right],\;\;D=\;\left[0\right]\;\;.$$

The system parameters and specifications of the pneumatic muscle actuators are presented in Table 2.

### 2.5. Controller Design

The control task is to design a controller that calculates an appropriate voltage signal *u*, which is sent to the proportional valve and provides control of the air-mass flow to the muscles, which then generate torque to rotate the manipulator arm. The control algorithm is implemented based on the state-feedback controller (linear quadratic controller—*LQR*). The *LQR* method is a powerful technique for controller design and aims to find optimal controller gains that minimize a cost function parameterized by two matrices, *Q* and *R*, that weight the state vector and the system input, respectively. The matrices *Q* and *R* are arbitrary, but the selection of these matrices is usually based on an iterative procedure based on experience and physical understanding of the control problem.

The finite-horizon continuous-time *LQR* is an optimal controller with state feedback based on minimizing the following performance criterion:(13)$$J_{LQR}=\int_{0}^{\infty }\left(x^{T}\;Q\;x\;+\;r\;u^{2}\right)\;dt\;\;\;\to \;\min$$
and addresses the optimal trade-off between tracking performance and control effort. The feedback control law that minimizes the value of the cost is:(14)u=−K x 
where the optimal controller gain matrix K is given as follows:(15)$$K=R^{-1}\;B^{T}\;H\;$$
and H is a symmetrical positive semi-definite square-matrix, which is found by solving the algebraic continuous-time Riccati differential equation:(16)$$H\;A+A^{T}\;H^{T}+Q-H\;B\;R^{-1}\;B^{T}\;H\;=0$$

The control law given in Equation (14) guarantees to maintain the system output as close as possible to the desired output with minimum control energy.

The weighting matrices *Q* and *R* are important components of an *LQR* optimization procedure. The composition of the *Q* and *R* elements has a critical impact on the performance of the control system. For simplicity, the off-diagonal elements of the *Q* matrix are zero. A reasonably simple choice for the *Q* and *R* matrices follows from Bryson’s rule, where the weighting matrices are selected as:(17)$$\begin{array}{c} Q_{ii}\;=\frac{1}{\max{\left(x_{i}\right)}^{2}}\;,\;\;\;\left(i=1,\ldots 5\right),\;\;\\ r=\frac{1}{\max{\left(u\right)}^{2}}\; \end{array}$$
where *x_i_* and *u* represent the acceptable (maximum) deviations of the corresponding state variables and the controller output from the equilibrium state. Determining the initial value of the matrix is often just the starting point for an iterative design procedure aimed at achieving the desired characteristics of the control loop. To obtain the diagonal values of the matrix *Q* for the selected values *x_i_* = [1 0.1 500,000] and the matrix *R* for the selected value *u* = 10, the following gain matrix *K* (using the Matlab function lqr.m) is determined for the optimal controller design:(18)$$K=\left[\;\;\begin{array}{ccc} 1 & 0.4325 & 0.0071 \end{array}\;\;\right]$$

This controller gain matrix *K* and the numerical values of the physical system parameters given in Table 2 will be used for the process control.

## 3. Results

A schematic diagram of the manipulator driven by PAMs in an antagonistic coupling is illustrated in Figure 9, while a photo of the laboratory equipment is given in Figure 10.

The hardware of the experimental system can be divided into two parts: the first part concerns the pneumatic system, which includes the PAM manipulator with pneumatic valve and the necessary measurement components, and the second part concerns the control system, which includes the control computer and the data acquisition module. The pneumatic part consists of an air-supply unit (SMC EAW 2000-F01), then a directly actuated proportional valve (Festo MPYE-5 1/8 HF-010B) and two pneumatic rubber muscles (Festo MAS -10-220N- AA-MC-K). The muscles are mounted antagonistically to actuate a swivel joint. Three pressure transducers (SMC ISE4-01-26) measure the air pressure inside the muscles and the air-supply pressure. The rotating torque is achieved by the pressure difference between two antagonistic muscles, and the lever with a pneumatic handle (Festo HGP-06-A) is rotated as a result. To measure the pulley angle *θ*, an industrial single-turn potentiometer (from Vishay Spectrol) attached to the rotating joint is used. The measured signals from the process are fed back to the control computer, which is equipped with a data acquisition card (NI DAQCard 6024E for PCMCIA with 12-bit A/D and 12-bit D/A converters).

The control software is implemented in the Matlab/Simulink environment using the Real-Time Workshop program (RTW) to generate ANSI C code from the block diagram processed in Simulink. The control signals calculated by the computer are sent to the proportional control valve via an electronic interface, then the air-mass flow rate through the valve can be regulated and the angular displacement of the manipulator arm can be controlled. By measuring the angular displacement of the shaft with the help of a rotary potentiometer, the control loop can be realized. The PAM is inflated by applying a control voltage to the proportional valve, which controls the flow of compressed gas into the cylindrical rubber hose. The controller regulates the air flow within the muscles to achieve a desired movement of the manipulator arm. A high-speed on/off valve (Matrix 758 series) is used to activate the gripper.

The experimental results for the angular displacement control of the single-joint manipulator arm actuated by PAMs are shown in Figure 11.

## 4. Discussion

The experimental results point out that the stable and well-damped response of the control system is obtained for both directions of the manipulator arm motion. In the lower part of the figure, the status of the gripper during the manipulator-arm movement is shown. When the arm is in the appropriate position, the gripper opens or closes by using an electromagnetic on/off valve, which allows the capture or release of the work pieces. The principle of operation of the practically realized manipulator driven by a pair of artificial pneumatic muscles can be seen in the Appendix A.

## 5. Conclusions

A single-degree-of-freedom manipulator actuated by PAMs has been designed, constructed and experimentally tested. Due to their properties, similar to biological muscles, fluidic muscles have a great potential in industrial applications and assembly automation for the actuation of robots and manipulators that enable some new features of the controlled system. Several varieties of possible solutions for the system design have been generated using a morphological matrix, and the chosen solution was practically realized. A simplified mathematical model of the system has been developed, which has been used in the controller synthesis procedure. Testing has been performed on the practically realized system for the task of transporting and sorting objects. The state-feedback controller has been utilized in the system control loop. The experimental results have shown the good dynamic behaviour of the closed-loop system. Based on these experiments, it could be concluded that PAMs are undoubtedly very suitable actuators for new types of industrial robots and manipulators. Their characteristics allow them to be easily assembled while improving performance compared to conventional pneumatic drives.

## Figures and Tables

**Figure 1 sensors-23-00776-f001:**
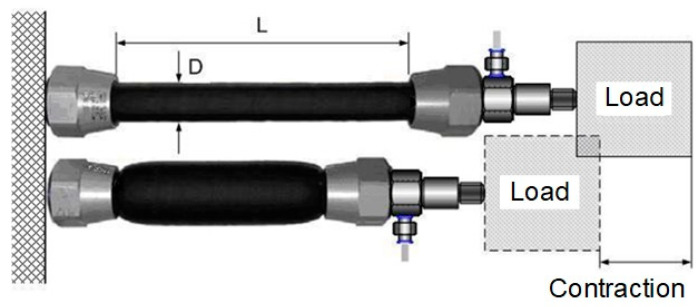
Operating principle of PAMs.

**Figure 2 sensors-23-00776-f002:**
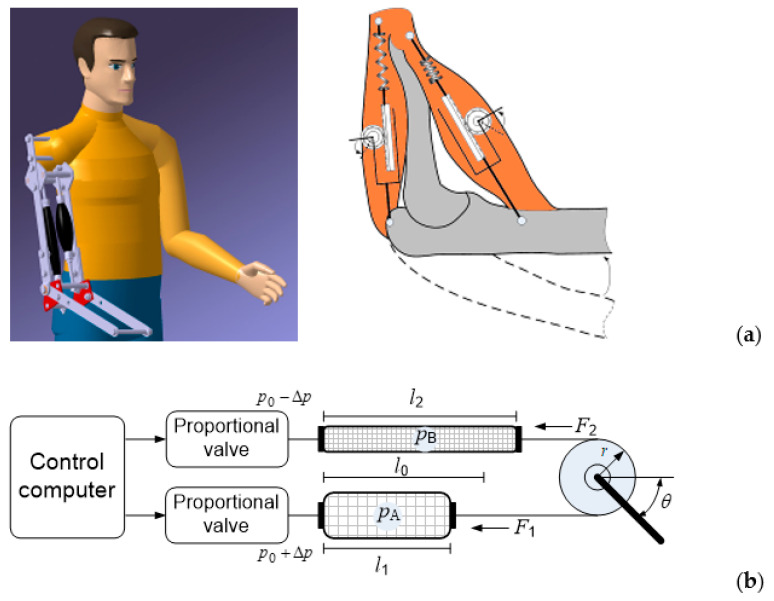
Similarity of working principle: (**a**) human arm, (**b**) manipulator.

**Figure 3 sensors-23-00776-f003:**
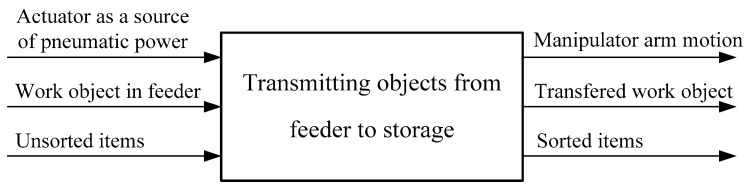
Functional structure of the system.

**Figure 4 sensors-23-00776-f004:**
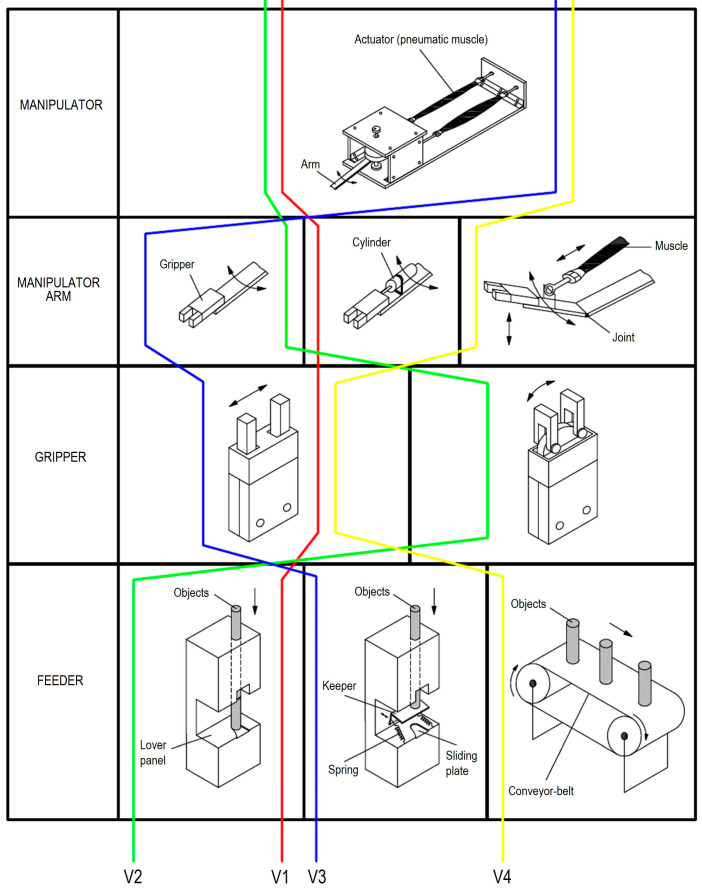
Morphological matrix with four considered variants (V1–V4) for manipulator construction.

**Figure 5 sensors-23-00776-f005:**
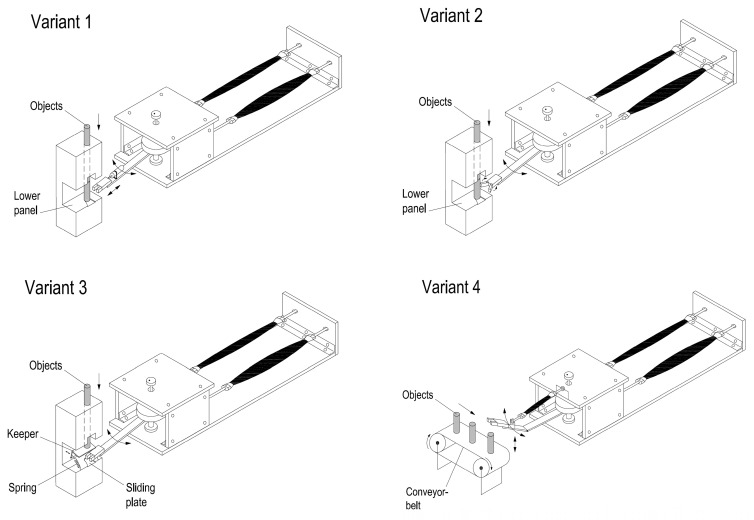
Four variants of system design.

**Figure 6 sensors-23-00776-f006:**
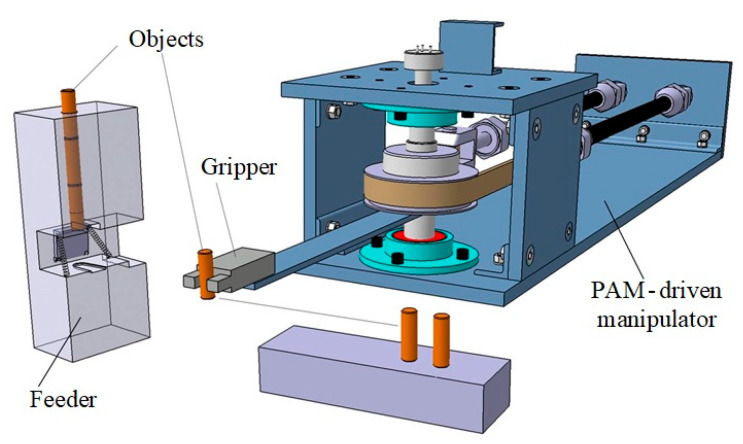
CAD model of the system for sorting tasks.

**Figure 7 sensors-23-00776-f007:**
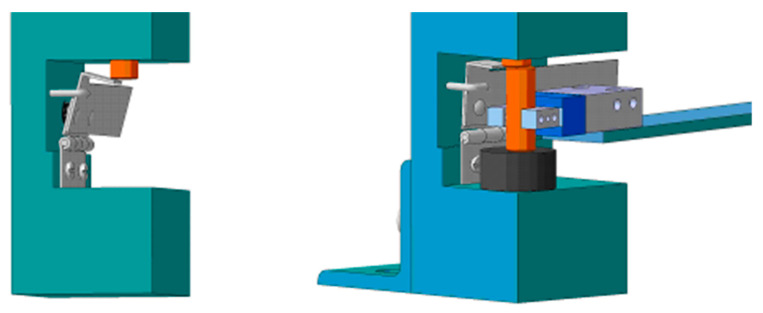
Work piece feeding and holding mechanism.

**Figure 8 sensors-23-00776-f008:**
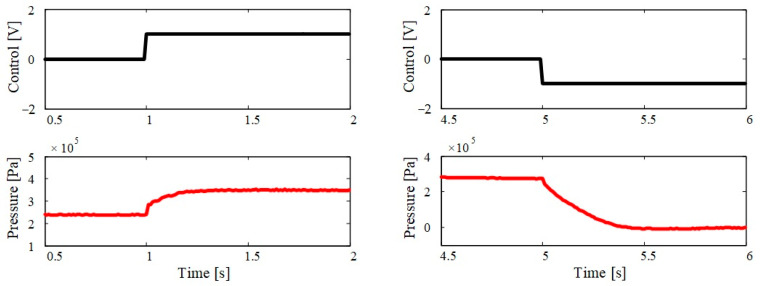
Transient response of antagonistic muscle pressures.

**Figure 9 sensors-23-00776-f009:**
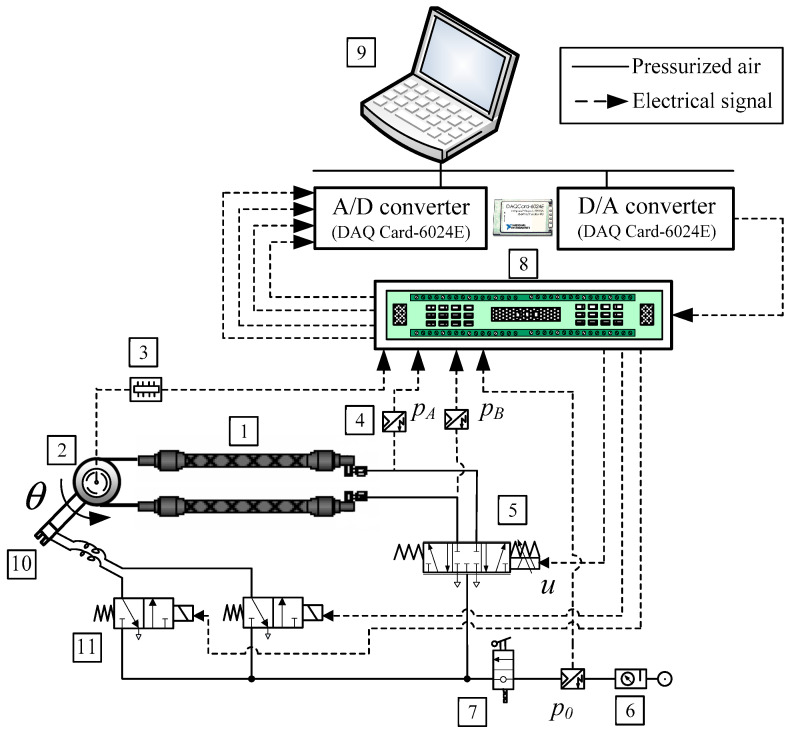
Schematic diagram of the control system.

**Figure 10 sensors-23-00776-f010:**
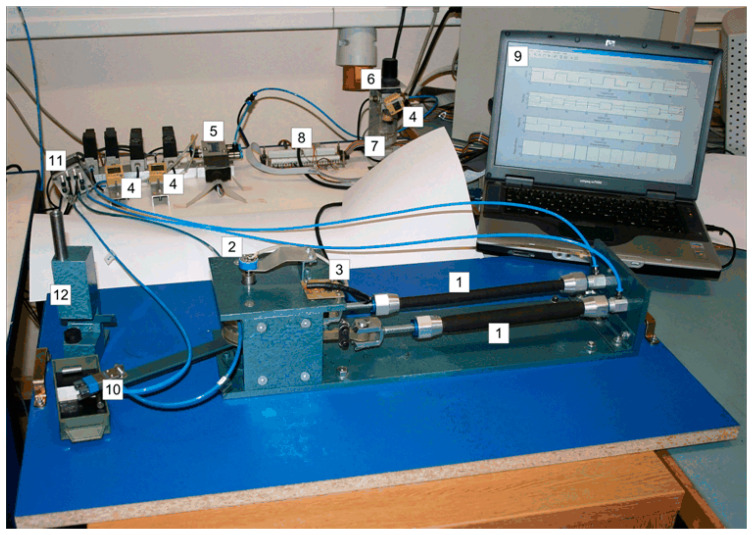
Photograph of the experimental equipment. Legend: 1—Pneumatic muscles, 2—Rotary potentiometer, 3—Voltage reference card, 4—Pressure sensors, 5—Proportional control valve, 6—Filter-regulator unit, 7—Manually operated valve, 8—Electronic interface, 9—Control computer with DAC card, 10—Gripper, 11—High-speed on/off solenoid valve, 12—Feeder.

**Figure 11 sensors-23-00776-f011:**
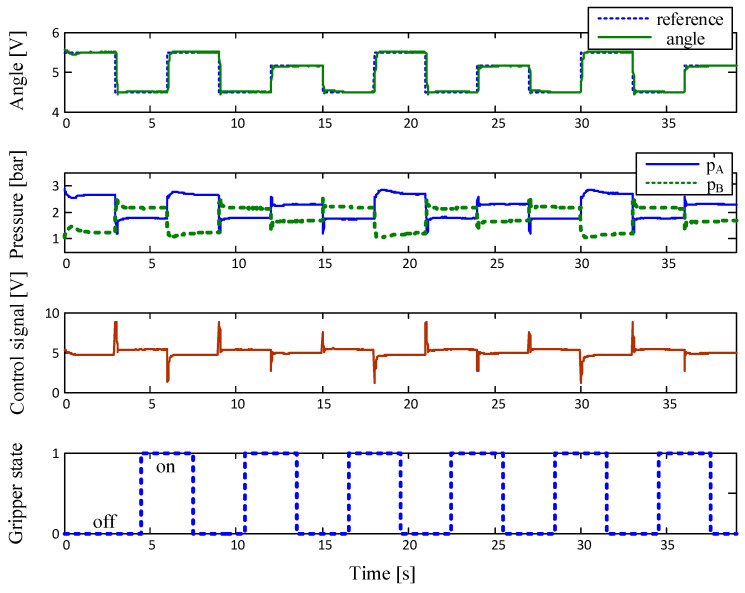
Experimental results.

**Table 1 sensors-23-00776-t001:** Comparison of different system designs.

Solution	Arm	Gripper	Feeder	Reliability	Total
	P→C	P→C	P→C	in Operation	
Variant 1	4→3	1→3	3→3	3	20
Variant 2	5→2	5→2	3→3	2	22
**Variant 3**	**1→1**	**1→2**	**4→4**	**3**	**16**
Variant 4	3→5	1→2	5→5	5	26

**Table 2 sensors-23-00776-t002:** Values of the system parameters.

System Parameters	Values
Nominal diameter of the muscle	$D_{0}\;=\;0.01\;m$
Initial angle of the braided shell	$\alpha_{0}\;=\;22\;\deg$
Initial contraction ratio of the muscle	$\varepsilon_{0}\;=\;0.1$
Radius of the pulley	r = 0.03 m
Fully contracted length of the muscle	$l_{\min}\;=\;0.176\;m$
Fully relaxed length of the muscle	$l_{\max}\;=\;0.22\;m$
Inertia of the mechanical rod	$I\;=\;2.25\;\times \;10^{-4}\;\;\mathrm{kg}\;m^{2}$
Supply pressure	$p_{s}=5.3\;\times \;10^{5}\;\;\mathrm{Pa}$
Forward gain of the pneumatic process	$K_{m}=1.1\;\times \;10^{5}\;\;\mathrm{Pa}/V$
Time constant of the pneumatic process	$\tau_{m}=0.05\;\;s$
Pressure in the muscle at the neutral valve position	$p_{0}=2.1\;\times \;10^{5}\;\;\mathrm{Pa}$

## Data Availability

Data sharing not applicable.

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
