# Peer review of "Design, Construction and Control of a Manipulator Driven by Pneumatic Artificial Muscles"

_sensors, 2023, doi:10.3390/s23020776_

Round 1

Reviewer 1 Report

There is not much innovative or new in this manuscript. Compared with the conference paper that the author has published, There is only a little improvement. Tearing a complete work into two parts and publishing them separately is not to be encouraged.

What I would like to see is a complete work, a thorough and systematic study of a problem.

The experimental data part of the article sparse. Please do the work more thoroughly and submit again.

Reviewer 2 Report

Dear Authors,
this is a sound piece of work. Unfortunately, it does not implement the requirements for the special issue.
"---This Special Issue aims to create a collection of papers that summarize the state of the art in advances to face the new robotics needs and complex system sense strategies. Formal and reliable approaches towards intelligent robotics should be tested in real applications or case studies. Experiences in machine learning, computer vision, or any other specific subject of computational intelligence tested in real or simulated robots will be considered."

Line                     Hint / Typo

92                        l0->l0

234                      Tm -> τm

281                      eq(17) check range for i

Round 2

Reviewer 1 Report

Compared with the previous version, it has been significantly improved. At present, the quality of this article is equal to the average quality of the publications in this journal. I agree to publish.

Reviewer 2 Report

Dear Authors,
thank you very much for the clarifications in the cover letter and the revision of the paper. Now the paper deserves to be published.